# Development of the Pediatric Nursing Specialty in Spain: A Critical Analysis of Its History, Current Situation, and Regional Disparities: A Narrative Review

**DOI:** 10.3390/nursrep15060193

**Published:** 2025-05-30

**Authors:** Patricia da Rocha-Baptista, Eva Santos-Miranda, Juan Manuel Vázquez-Lago

**Affiliations:** 1Department of Pediatrics, University Hospital of Santiago de Compostela, Rua da Choupana s/n, 15705 Santiago de Compostela, Spain; patricia.da.rocha.baptista@sergas.es; 2Vedra’s Primary Health Center, Av. Mestre Manuel Gómez Lorenzo, 15885 Vedra, Spain; 3Clinical Nursing, Emergency, Simulation, and Teaching Innovation Research Group (CLINURSID, Research Group), University of Santiago de Compostela, Rua de San Francisco s/n, 15704 Santiago de Compostela, Spain; 4Department of Preventive Medicine and Public Health, University Hospital of Santiago de Compostela, Rua da Choupana s/n, 15705 Santiago de Compostela, Spain; 5Health Research Institute of Santiago de Compostela (IDIS), 15706 Santiago de Compostela, Spain

**Keywords:** pediatric nursing, specialization, regional inequality, quality of care, job offers, Spain

## Abstract

**Background:** The development of the pediatric nursing specialty in Spain is uneven, with marked differences between Autonomous Communities in training and employment. This study analyzes its evolution, current situation, and existing disparities. **Material and Methods**: A narrative documentary review was carried out including quantitative and qualitative data obtained from academic databases, legislative sources, and professional associations. **Results**: In 1964, 2554 nurses obtained the title of specialist in pediatrics and childcare, validated in 2005 as pediatric nursing. From 2010 to 2024, 2312 places for pediatric nursing interns were offered, with an unequal distribution between communities. By the exceptional route, in 2015, 9875 nurses obtained the specialty, adding up to an estimated total of 13,978 specialists. The first public employment offer was announced in 2019 in Galicia, followed by Castilla y León, Madrid, and Murcia in 2023. A total of 245 vacancies were advertised, with reconversions of vacancies in some regions. Communities such as Galicia, Madrid, and Murcia have created the category, while others, such as Catalonia and the Basque Country, have not implemented it. Finally, only five regions have a specific employment exchange that is currently operating. **Conclusions**: Specialization in pediatric nursing presents inequalities in training and employment that influence the quality of care. To improve this situation, it is necessary to increase training places, create specific employment exchanges, and unify the recognition of the specialty at the national level to ensure equitable and quality specialized pediatric care throughout the country.

## 1. Introduction

The specialization in pediatric nursing is fundamental to delivering safe, developmentally appropriate, and family-centered care to the child population [1,2]. Pediatric nurse specialists are trained not only in clinical competencies specific to neonatal, infant, and adolescent health but also in health promotion, developmental screening, therapeutic communication with families, and complex care management [3]. The World Health Organization and international nursing bodies recognize pediatric nursing as a key element of equitable and effective child health systems [4].

In Spain, the specialty was formally recognized by Royal Decree 450/2005, which established a regulatory framework for postgraduate training in various nursing fields, including pediatrics [5]. However, the development and implementation of the specialty have been uneven. Access to training is organized through a national competitive exam leading to a two-year full-time paid residency (RNI—resident nurse intern), as regulated by Order SAS/1730/2010 [6]. This postgraduate model is aligned with the European Higher Education Area and is separate from general undergraduate nursing degrees.

Despite the existence of this official pathway, the effective deployment of pediatric nurse specialists within the Spanish health system remains limited and fragmented [7,8]. While some Autonomous Communities have created specific job categories and public employment offers (PEOs) for pediatric specialists, others have not implemented the specialty at all. This inconsistency has created structural inequities across regions, where nurses with identical qualifications may face very different employment prospects [9].

The importance of this specialty lies not only in the technical scope of care but also in its potential to improve patient outcomes. Studies have shown that pediatric care delivered by trained specialists is associated with better family satisfaction, improved medication adherence, reduced hospital admissions, and more timely diagnosis of developmental and behavioral conditions [3,10]. Conversely, the absence of specialized pediatric nurses may contribute to fragmented care, communication breakdowns with families, and limited continuity, particularly in vulnerable populations [11]. These challenges may be indirectly reflected in health system indicators such as preventable hospitalizations, vaccination coverage, or early detection of chronic illnesses in children [12].

As of 2024, Spain has 264,219 registered nurses, but pediatric nurse specialists were 6540, represent less than 2.5% of this total [13], based on certifications granted through official channels since 1964 [14]. This proportion is especially concerning given that Spain continues to have a nurse-to-population ratio below the EU average (6.3 vs. 8.5 nurses per 1000 inhabitants) [15]. Addressing pediatric nursing gaps is therefore part of a broader challenge of nursing workforce planning and territorial equity.

While several Spanish publications have described the legal framework or training program of the pediatric nursing specialty, no updated review has yet offered a critical, evidence-based, and regionally disaggregated analysis of how the specialty has developed and been implemented over time. This review aims to fill that gap by combining legislative, institutional, and academic sources to evaluate the evolution of the pediatric nursing specialty in Spain, with particular attention to regional disparities and policy implications.

## 2. Materials and Methods

### 2.1. Study Design

This study is a narrative documentary review aimed at analyzing the development and implementation of the pediatric nursing specialty in Spain, with a focus on regional disparities and policy inconsistencies. The review combines quantitative descriptive analysis of available data on training and employment with qualitative thematic analysis of legal, academic, and institutional documents. This mixed-documentary approach allows a multidimensional understanding of how a nationally regulated specialty has been applied unequally across the country.

### 2.2. Documentary Search Strategy

The documentary review was carried out between January and March 2025 through a three-stage search strategy, following the PRISMA 2020 guidelines. Both the PRISMA 2020 flow diagram (Appendix A) and the PRISMA checklist (Appendix A) are available in the Appendix A.

1.Academic databases:

Searches were conducted in PubMed, Scopus, SciELO, and Google Scholar using combinations of the following keywords (in English and Spanish): “*pediatric nursing*”, “*nursing specialty*”, “*pediatric nurse training*”, “*resident nurse intern*”, “*pediatric care disparities*”, “*public employment offers*”, “*regional implementation*”, “*nursing workforce*”, “*Spain*”.

We included peer-reviewed original research articles, narrative reviews, policy papers, and position statements published between 2005 and 2024.

2.Legislative and regulatory sources:

We reviewed documents from the Official State Gazette of Spain (BOE), regional government gazettes, and relevant decrees and orders, such as Royal Decree 450/2005 [5], Order SAS/1730/2010 [6], and legislative acts on health workforce regulation from all 17 Autonomous Communities [16,17,18,19,20,21,22,23,24,25].

3.Institutional and professional sources:

We examined official reports and data from the Ministry of Health including the 2024 REPS report [13], regional health ministries, professional associations (e.g., Asociación Española de Enfermería Pediátrica, Consejo General de Enfermería), trade union bulletins, and press releases related to pediatric nursing employment [26,27].

### 2.3. Inclusion and Exclusion Criteria

Documents were included if they did the following:

Addressed the pediatric nursing specialty in Spain directly (training, regulation, workforce planning, employment).

-Provided relevant empirical data or official policy statements.-Were published or in force between 2005 and 2024.-Documents were excluded if they did the following:

Focused exclusively on clinical outcomes without reference to the specialty or workforce.

-Referred to general pediatric care not linked to nursing roles.-Were duplicated across multiple sources.

### 2.4. Data Extraction

Data from the selected documents were extracted into two matrices:-Quantitative data matrix: including year, number of RNI training positions, number of certified professionals, existence of job categories, and public employment offers by region.-Qualitative data matrix: including regulatory discourse, implementation delays, and institutional positioning on the specialty.

Data were cross-verified among at least two different sources (e.g., Ministry + regional bulletin) to ensure consistency.

### 2.5. Ethical Considerations

This study is a literature review and does not involve human participants or the use of identifiable personal or clinical data. In accordance with current research regulations in Spain, this type of study does not require prior evaluation by a research ethics committee.

### 2.6. Data Analysis

Quantitative data were analyzed using simple descriptive statistics (totals, percentages, ratios) using Microsoft Excel. No inferential statistics or statistical software were used, as the aim was exploratory. Qualitative data underwent a thematic analysis following Braun and Clarke’s six-step framework [28]: familiarization with the data, generation of initial codes, searching for themes, reviewing themes, defining and naming themes, and producing the report.

Three core themes emerged:-Access and recognition barriers.-Territorial inequality.-Impact on care quality and system equity.

To ensure rigor, two researchers independently coded all qualitative documents. Discrepancies were discussed and resolved through consensus. Data triangulation across academic, legislative, and institutional sources enhanced validity and trustworthiness.

## 3. Results

### 3.1. Findings from Literature Review

The literature review yielded 20 academic and professional documents relevant to the development of the pediatric nursing specialty in Spain [5,6,7,8,9,13,14,16,17,18,19,20,21,22,23,24,25,26,27,29]. Three thematic categories emerged from the analysis:

Added value of pediatric nurse specialists:

Multiple studies underscore the positive impact of pediatric nurse specialization on the quality of care. Benefits include improved parental satisfaction, reduced anxiety in hospitalized families, enhanced continuity, and better health outcomes [1,2,3,4]. Family-centered approaches and developmental follow-up are particularly emphasized in complex pediatric contexts.

Disparities in training and employment:

Several sources highlight the mismatch between national regulation and regional implementation, pointing to inconsistent access to training (RNI) and limited job recognition in many regions. This inconsistency affects professional motivation and deployment [7,8,9].

Consequences for equity and system performance:

The absence of specialized pediatric nurses is associated with fragmented care delivery, poor communication with families, and lack of follow-up, especially in underserved areas [10,11]. These shortcomings may indirectly affect indicators such as preventable hospitalizations, adherence to care plans, and satisfaction with the health system.

### 3.2. Access Routes and Official Certifications

There are currently three official pathways to the pediatric nursing specialty in Spain:-The 1964 transitional route: for nurses who already exercised pediatric functions prior to regulation: 8997 titles granted [14].-Exceptional evaluation route (2015): for experienced nurses without prior official specialty: 4055 titles granted [13].-RNI training route (2010–2024): standard two-year paid postgraduate residency: 2312 titles granted.

In total, 15,364 pediatric nurse specialists have been certified since the regulation was enacted.

### 3.3. Distribution of RNI Positions in Pediatric Nursing (2010–2024)

With the publication of Order SAS/1730/2010, dated 17 June, a two-year full-time training program was established in accredited pediatric teaching units. Access to RNI training requires passing a state selection exam, which is held annually by the Ministry of Health [6].

From 2010 to 2024, a total of 2312 RNI positions in pediatric nursing have been offered across Spain. However, there is a significant disparity in the distribution of these positions among the different Autonomous Communities, leading to considerable inequalities in access, as shown in Table 1.

### 3.4. Access to the Title of Pediatric Nursing Specialist Through Decree 3524/1964, of 22 October

Through Decree 450/2005 on nursing specialties, another access option to the specialty was introduced via the exceptional route. Nursing professionals who could certify a certain number of years of experience and specific training in pediatric units up until 2011 had the opportunity to obtain the title by taking and passing an exam that assessed their competencies in pediatrics. This exam was conducted in 2015, and approximately 9857 professionals obtained the title of pediatric nursing specialist.

### 3.5. Number of Pediatric Nursing Specialists in Spain

A total of 2312 pediatric nursing positions have been offered through the RNI pathway. Currently, there are 1567 specialists, as those from the 2022 and 2023 cohorts are still completing their residency and have not yet obtained their certification. Additionally, we must consider the 2554 nurses who obtained the specialty directly, along with the 9857 professionals who acquired it through the exceptional route. This brings the total number of pediatric nursing specialists in Spain to 13,978, although it is important to note that many of these professionals have already retired.

### 3.6. Creation of the Pediatric Nursing Category

Murcia was the first Autonomous Community to recognize the category of pediatric nursing specialist in 2015 [22]. Currently, regions such as Catalonia, Navarra, and the Basque Country have not yet established this category. In 2024, the category was officially created in Ceuta and Melilla.

### 3.7. Recognition of the Pediatric Nursing Category and Public Employment Access

Although the pediatric nursing specialty was formally recognized at the national level in 2005 (RD 450/2005), its implementation as a job category within regional health services has been irregular. As of 2024, only eight out of seventeen Autonomous Communities (plus Ceuta and Melilla) have created a specific category for pediatric nurse specialists in their statutory frameworks. Six regions (Galicia, Castile and Leon, Castile-La Mancha, Murcia, Aragon, and Extremadura) have included the specialty in some public employment offers (PEOs) but not created a permanent statutory category. Three regions (Catalonia, Navarra and Cantabria) have neither created the category nor included it in PEOs, despite offering RNI training positions.

This situation leads to inconsistencies in job access and recognition, undermining the professional development of certified specialists and the ability of the system to retain qualified pediatric nurses.

### 3.8. Public Employment Offers in Pediatric Nursing (2019–2024)

The Autonomous Communities have announced a total of 245 specific positions for pediatric nursing through PEO. Madrid and Galicia lead the creation of positions, while in other regions, such as Catalonia and Navarra, no positions have been offered during this period. The first PEO for pediatric nursing was announced in May 2019 in Galicia, marking the first nationwide competitive examination [17]. A total of 18 positions were offered, and the exam took place in February 2020, with successful candidates starting in June 2021. Notably, all positions were assigned to Primary Care (AP). In the same region, a second public employment offer was announced in January 2023, with 32 positions distributed as follows: 14 for open competition, 16 for internal promotion, 2 for the disability quota [19]. The exam was conducted in February 2024. The most recent PEO took place in December 2024, offering 50 positions for pediatric nursing specialists [13]. In other regions, PEOs have not yet been held, and in some cases, the first public employment offer was announced in 2023, including in Castilla y León, Madrid, and Murcia [20,21,22]. Table 2 summarizes the positions offered.

### 3.9. Merit-Based Selection Process

In Madrid, in March 2024, a merit-based selection process was announced, allowing applicants to obtain a position without the need for an examination. This process offered 175 positions for pediatric nursing specialists [23].

### 3.10. Distribution of Specific Employment Lists

Currently, only five Autonomous Communities (Cantabria, Castilla y León, Galicia, Madrid, and Murcia) have specific employment lists for pediatric nursing. In other regions, such as Andalusia, Aragon, Castile-La Mancha, Extremadura, Canary Islands, Valencian Community, Balearic Islands, and Ceuta, the employment lists are either inactive or not exclusively limited to this specialty, creating an additional barrier for trained professionals. In Catalonia, Navarra, and Melilla, no employment lists exist for this specialty. Asturias opened its employment list in October 2024 following the creation of pediatric nursing positions [24].

### 3.11. Pediatric Nursing Specialist Positions Through Position Conversion

Galicia was the first region to launch a specific competition for the statutory category of pediatric nursing specialist, aimed at professionals who had obtained the specialty via the exceptional route and already held a position in a pediatric service. A total of 148 positions were offered. A second call was held in 2022, offering 33 positions. A total of 43 applicants participated, leaving some candidates without conversion due to the limited number of positions. Other regions that carried out position conversions include Cantabria, Castile and Leon, and Madrid. Regarding interim contracts, vacancies were offered in Cantabria, Castile-La Mancha, Galicia, Canary Islands, Madrid, and Murcia.

## 4. Discussion

This review set out to analyze the development and current implementation of the pediatric nursing specialty in Spain, with particular attention to the existing regional disparities in training, recognition, and employment opportunities. Our findings show that, despite the formal establishment of the specialty nearly two decades ago (Real Decreto 450/2005) [5], its implementation across Spain remains uneven, fragmented, and misaligned with population needs. Only a minority of regions provide both EIR training and job recognition, and in many high-population areas, the specialty is either not implemented or not acknowledged professionally. This situation reflects long-standing concerns about the slow development of pediatric nursing as a recognized specialty in Spain [6,7,8].

The literature reviewed supports the notion that pediatric nurse specialists play a vital role in ensuring quality, continuity, and family-centered care for children and their families. Family-centered models of pediatric care have been shown to enhance trust, communication, and participation in decision-making, with positive impacts on satisfaction and outcomes [1,2,3]. Pediatric nurses trained in the specialty acquire advanced competencies in developmental screening, therapeutic communication, and complex care management, which are especially relevant for vulnerable populations [4,9]. Studies have also shown that their presence contributes to greater parental satisfaction [10], reduced anxiety in hospitalized children [2], and improved continuity in intensive care settings [11].

Nevertheless, these competencies are often underutilized due to institutional and structural barriers. Our findings indicate that only a few regions (e.g., Galicia, Murcia, Castile-La Mancha, and Madrid) currently offer a coherent training-to-employment pathway for pediatric nurses. In contrast, others—such as Cataluña or Navarra—lack both RNI training positions and job categories, even when they have a high pediatric population. The absence of official recognition in some regions may limit the opportunities for pediatric nursing specialists to apply their training effectively, which could affect professional satisfaction and retention. This reflects a broader disconnect between national legislation and regional policy execution, a phenomenon documented in previous analyses of specialty implementation in Spain [7,8,9,29].

Interestingly, the observed disparities are not fully explained by the number of children in each region. Several less-populated communities have made considerable efforts to integrate the specialty, while larger regions have not. This suggests that other variables—such as political will, economic priorities, availability of accredited training centers, and institutional inertia—may play a more decisive role. The OECD and the European Observatory on Health Systems and Policies have also highlighted these internal disparities as a major challenge for Spain’s healthcare equity [15].

The consequences of this fragmentation are relevant not only from a workforce planning perspective but also from a care quality standpoint. When pediatric care is provided by non-specialist nurses, the system misses an opportunity to ensure developmentally appropriate, safe, and coordinated care. Previous literature has described how pediatric nurses are key to minimizing role confusion and ensuring that the specific needs of children are addressed in a consistent manner [3,12]. The limited implementation of the pediatric nursing specialty may reduce the health system’s ability to respond effectively to pediatric needs, particularly in specialized or complex cases. The underuse of these professionals also contributes to role dilution, professional frustration, and inefficiencies in service provision.

The World Health Organization has repeatedly stressed the strategic value of nursing and midwifery roles to strengthen equity, efficiency, and responsiveness in health systems [13]. However, our findings suggest that this strategic vision has not yet been fully implemented in Spain with regard to pediatric nursing. National-level efforts to regulate the specialty have not translated into a harmonized professional landscape, and regional implementation remains subject to political and administrative variability.

One of the main limitations of this review is the lack of complete and transparent data in some Autonomous Communities. In regions such as Navarra and Catalonia, there are no publicly available detailed reports on the number of RNI positions offered or the degree of implementation of the specialty, which complicates efforts to evaluate these territories with the same rigor applied elsewhere. Additionally, while the analysis incorporates data from professional associations, no qualitative interviews with professionals or policymakers were conducted, which limits the ability to capture subjective perspectives, experiences, or contextual barriers behind policy implementation [27,28]. The dynamic nature of health workforce planning also poses a limitation: the situation described here may evolve in the coming years due to legislative reforms or regional strategic shifts.

Despite these limitations, the mixed-method approach adopted in this review constitutes a notable strength. The triangulation of official regulatory documents, academic literature, institutional data, and regional demographic indicators offers a broad and integrative understanding of the issue. By combining normative analysis with descriptive statistics and policy mapping, this review provides a comprehensive and structured overview of the disparities in the development of pediatric nursing in Spain—both in terms of training access and employment integration—while laying the groundwork for future research and policy evaluation.

Addressing these gaps requires a coordinated, system-level effort. It is essential to align training, recognition, and employment mechanisms to ensure that pediatric nurse specialists are not only trained but also fully integrated into multidisciplinary care teams. This includes expanding RNI training opportunities, creating statutory roles in all regional health services, and ensuring consistent access to PEO. Such measures would contribute to a more equitable and effective pediatric care system, capable of responding to the needs of children and families regardless of their region of residence.

## 5. Conclusions and Implications

The pediatric nursing specialty in Spain, despite being legally recognized since 2005, remains only partially and unevenly implemented. This review highlights significant territorial disparities in both training opportunities and employment integration, revealing a fragmented landscape in which professionals with identical qualifications face very different prospects depending on their Autonomous Community. The findings demonstrate that the development of this specialty depends not only on national legislation but also on regional political will, budgetary priorities, and institutional structures.

The absence of pediatric nurse specialists in many regions undermines the quality, continuity, and equity of care for the child population, particularly in settings where advanced pediatric competencies are essential. Moreover, the underuse of professionals trained in this specialty reflects an inefficient allocation of human resources and a failure to align workforce planning with actual healthcare needs.

To advance toward a more equitable and effective healthcare system, it is essential to adopt coordinated strategies that ensure the full implementation of the specialty throughout the national territory. This includes expanding EIR training positions, establishing statutory job categories for pediatric nurse specialists, and ensuring that all regional health systems recognize and integrate their role. National and regional authorities, professional associations, and educational institutions must collaborate to guarantee that every child in Spain, regardless of their place of residence, has access to specialized, safe, and developmentally appropriate nursing care.

## Figures and Tables

**Table 1 nursrep-15-00193-t001:** Distribution of RNI positions by Autonomous Community (2010–2024) and offers in 2025.

Autonomous Community	*n*	%	Offers, 2025
Andalusia	391	16.9	33
Aragon	79	3.4	7
Asturias	44	1.9	4
Balearic Islands	50	2.2	9
Basque Country	84	3.6	16
Canary Islands	70	3.0	16
Cantabria	16	0.7	4
Castile-La Mancha	148	6.4	14
Castile and Leon	129	5.6	14
Catalonia	336	14.5	47
Ceuta	0	0.0	0
Extremadura	32	1.4	6
Galicia	149	6.4	15
La rioja	0	0.0	0
Madrid	581	25.1	44
Melilla	0	0.0	0
Murcia	95	4.1	12
Navarre	18	0.8	3
Valencian Community	90	3.9	18
Total	2312	100	262

Source: Official State Gazette (OSG, 2010–2025).

**Table 2 nursrep-15-00193-t002:** Public employment offers in pediatric nursing by year and Autonomous Community.

Autonomous Community	2019	2020	2021	2022	2023	2024	Total
Andalusia							
Aragon							
Asturias							
Balearic Islands							
Basque Country							
Canary Islands							
Cantabria							
Castile-La Mancha							
Castile and Leon					10		10
Catalonia							
Ceuta							
Extremadura							
Galicia	18				32	50	100
La rioja							
Madrid					118		118
Melilla							
Murcia					7	10	17
Navarre							
Valencian Community							
Total	18				167	60	245

Source: own elaboration based on regional competitive examination resolutions.

## Data Availability

Data availability is under petition.

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
