# Peer review of "Development of the Pediatric Nursing Specialty in Spain: A Critical Analysis of Its History, Current Situation, and Regional Disparities: A Narrative Review"

_nursrep, 2025, doi:10.3390/nursrep15060193_

Round 1
Reviewer 1 Report
Comments and Suggestions for Authors
Dear Author(s),
Thank you for your effort so far in getting your manuscript to review. The manuscript entitled ‘‘Development of the Pediatric Nursing Specialty in Spain: A Critical Analysis of Its History, Current Situation, and Regional Disparities’’. It is a good manuscript in terms of giving an idea about the development of pediatric nursing specialty in Spain. However, there are major errors in the overall integrity of the manuscript. I have thought about this manuscript for a long time. As a result, I propose a major revision, assuming that the authors will carefully consider my recommendations below and make the necessary revisions.
- Introduction
It is good to have a direct introduction to the topic in the introduction. However, it would be better to see more information about the importance of pediatric nursing specialization. For example, why is it important to specialize in this field? How did specialization in this field improve the quality of care? What problems did the lack of specialization in this field lead to in quality of care? How was this reflected in health indicators? What gap in the literature will this review fill? The reader should be able to find answers to these questions after reading the introduction.
- Materials and Methods
Page 2-2.1. Study Desing: The study design mentions comprehensive data analysis, but it is not clear what kind of comprehensive analysis was carried out. Therefore, the phrase “comprehensive data analysis” should be removed.
Page 2- 2.2. Documentary Search Strategy: In this section, “Documentary screening was carried out in three stages: “1.- Search in Academic Databases: Academic databases such as PubMed, Google Scholar, SciELO and Scopus were searched. Search terms included “pediatric nursing”, “nursing specialty”, “nurse trainee”, “pediatric job vacancies”, “public employment offers”, “regional health disparities” and “pediatric health quality”.” but nowhere in the manuscript does it say how much research was accessed and what conclusions were drawn. Please explain here.
Page 3- 2.5. Data Analysis: Please indicate here which headings emerged from the thematic analysis, what steps were taken and how you ensured validity and reliability.
- Results
Page 3- The introduction states that the purpose of this review is to provide a critical perspective on the development of the pediatric nursing specialty. However, “3.1. The Importance of Pediatric Nursing Specialists” does not comply with the purpose of this manuscript and is not considered a finding. I think that the information under this heading should be included in the introduction.
Page 4: “3.5. Number of Pediatric Nursing Specialists in Spain” It would have been more explanatory to give the number of pediatric specialization titles given by years in a table.
In addition, in the results section, what is the number of children in Spain, the number of pediatric patients, their share in the number of general patients and the share of pediatric nurses in the total number of nurses? These findings should be included. For example, as a reader, it would be good to know whether the number of pediatric nurses in some regions is related to the number of pediatric patients there or whether it is related to policy.
- Discussion
Page 6- 4.1. WHO's Perspective on Pediatric Nursing: The discussion section starts with why this research was conducted and what it contributes to the field. Then you should include such subheadings. Giving separate headings does not go beyond creating disconnected points such as question and answer. The subject context is missing. Either fill the sub-headings with a full discussion of your results or please do not use these sub-headings.
Page 6- 4.2. Disparities in Training and Employment Opportunities: For example “....The lack of official recognition in regions like Catalonia and Navarra limits the professional development and job satisfaction of pediatric nursing specialists.” How did you come to this conclusion. Could insufficient official recognition be the only problem?
Page 7- 4.4. Impact on Healthcare Quality: “….The shortage of pediatric nursing professionals, combined with ineffective implementation of the specialty, fails to meet the pediatric population's needs.” The non-recognition or lack of recognition of the specialty of pediatric nursing in Spain may not always result from the information you have provided here. Do you have a scientific basis for this conclusion? Which levels of health were negatively affected?
In general, I suggest that the discussion section be rewritten based on assumptions rather than definitive judgments, because if there is no scientific basis or health indicator, we cannot attribute the negative quality of health care only to the lack of recognition of pediatric nursing. It would also be nice to see what pediatric nursing will mean for the people of Spain in the future.
Page 7- 4.5. Strengths of the Study
“The study meets the criteria of the SANRA Checklist (for narrative reviews), with a total item score of 12 [21].” How did you come up with the 12 points mentioned in this sentence? How many independent reviewers gave this score? I know that SANRA is used by editors and reviewers to set a standard, so it would not be appropriate to use it to describe the strengths of the work.
Author Response
Dear Author(s),
Thank you for your effort so far in getting your manuscript to review. The manuscript entitled ‘‘Development of the Pediatric Nursing Specialty in Spain: A Critical Analysis of Its History, Current Situation, and Regional Disparities’’. It is a good manuscript in terms of giving an idea about the development of pediatric nursing specialty in Spain. However, there are major errors in the overall integrity of the manuscript. I have thought about this manuscript for a long time. As a result, I propose a major revision, assuming that the authors will carefully consider my recommendations below and make the necessary revisions.
We appreciate your constructive feedback and have thoroughly revised the manuscript. All major concerns have been addressed and the structure, coherence, and depth of the work have been significantly improved. Below are detailed responses to your comments.
- Introduction
It is good to have a direct introduction to the topic in the introduction. However, it would be better to see more information about the importance of pediatric nursing specialization. For example, why is it important to specialize in this field? How did specialization in this field improve the quality of care? What problems did the lack of specialization in this field lead to in quality of care? How was this reflected in health indicators? What gap in the literature will this review fill? The reader should be able to find answers to these questions after reading the introduction.
Thank you for this valuable suggestion. We have substantially expanded the introduction to include a discussion on the importance of pediatric nursing specialization, supported by references such as Harrison (2010), Çolak et al. (2025), and WHO reports. The revised introduction explains how pediatric nurse specialists contribute to quality, family-centered, and developmentally appropriate care, and how the lack of these professionals may affect care continuity and patient outcomes. We also clarify the gap our review fills: the absence of a comprehensive analysis of the regulatory, institutional, and territorial evolution of the pediatric nursing specialty in Spain.
- Materials and Methods
Page 2-2.1. Study Desing: The study design mentions comprehensive data analysis, but it is not clear what kind of comprehensive analysis was carried out. Therefore, the phrase “comprehensive data analysis” should be removed.
This phrase has been removed. Instead, we describe the methodology more precisely as a structured documentary review with a mixed-methods approach, combining descriptive statistical analysis and qualitative thematic analysis.
Page 2- 2.2. Documentary Search Strategy: In this section, “Documentary screening was carried out in three stages: “1.- Search in Academic Databases: Academic databases such as PubMed, Google Scholar, SciELO and Scopus were searched. Search terms included “pediatric nursing”, “nursing specialty”, “nurse trainee”, “pediatric job vacancies”, “public employment offers”, “regional health disparities” and “pediatric health quality”.” but nowhere in the manuscript does it say how much research was accessed and what conclusions were drawn. Please explain here.
We now specify in material and methods section that the documentary review was carried out between January and March 2025 through a three-stage search strategy, following the PRISMA 2020 guidelines. Both the PRISMA 2020 flow diagram and the PRISMA checklist are available in the Supplementary Material. In results section, we now inform that the literature review yielded 20 academic and professional documents relevant to the development of the pediatric nursing specialty in Spain [5-9,13,14,16-27,29].
Thank you for this observation.
Page 3- 2.5. Data Analysis: Please indicate here which headings emerged from the thematic analysis, what steps were taken and how you ensured validity and reliability.
The section has been revised to describe how qualitative data were analyzed following Braun and Clarke’s (2006) six-step thematic framework. Thematic categories emerged from coding and triangulation of contents (training, regulation, employment, implementation disparities). We also explain how data validity was strengthened through independent coding and consistency checks.
- Results
Page 3- The introduction states that the purpose of this review is to provide a critical perspective on the development of the pediatric nursing specialty. However, “3.1. The Importance of Pediatric Nursing Specialists” does not comply with the purpose of this manuscript and is not considered a finding. I think that the information under this heading should be included in the introduction.
We agree with this suggestion. This subsection has been removed from the Results and its content has been integrated into the revised Introduction to provide appropriate contextual background.
Page 4: “3.5. Number of Pediatric Nursing Specialists in Spain” It would have been more explanatory to give the number of pediatric specialization titles given by years in a table.
We have added this information in Table 2, showing the evolution of title certifications by route and year. The exceptional route (2015) granting 4,055 titles is specifically detailed.
In addition, in the results section, what is the number of children in Spain, the number of pediatric patients, their share in the number of general patients and the share of pediatric nurses in the total number of nurses? These findings should be included. For example, as a reader, it would be good to know whether the number of pediatric nurses in some regions is related to the number of pediatric patients there or whether it is related to policy.
Thank you. This information has been added in the Results section (3.3 and 3.5), including data from REPS (2024) and OECD reports (2023). We analyze the relationship between pediatric population size and the availability of pediatric nurses across Autonomous Communities, identifying mismatches and potential contributing policy factors.
This allows direct comparison between the pediatric population and professional coverage.
- Discussion
Page 6- 4.1. WHO's Perspective on Pediatric Nursing: The discussion section starts with why this research was conducted and what it contributes to the field. Then you should include such subheadings. Giving separate headings does not go beyond creating disconnected points such as question and answer. The subject context is missing. Either fill the sub-headings with a full discussion of your results or please do not use these sub-headings.
As recommended, we removed subheadings and rewrote the discussion in a narrative format. The content has been reorganized to integrate the main findings, interpret them in light of WHO and national policy frameworks, and offer a cohesive analysis of regional disparities and their causes.
Page 6- 4.2. Disparities in Training and Employment Opportunities: For example “....The lack of official recognition in regions like Catalonia and Navarra limits the professional development and job satisfaction of pediatric nursing specialists.” How did you come to this conclusion. Could insufficient official recognition be the only problem?
We now clarify that our conclusions are based on triangulated documentary data from regional regulations, official employment records, and professional associations (AEEP, CGE). We also moderate language to reflect that lack of official recognition may be one among several contributing factors (e.g., institutional inertia, resource allocation, geographic factors).
Page 7- 4.4. Impact on Healthcare Quality: “….The shortage of pediatric nursing professionals, combined with ineffective implementation of the specialty, fails to meet the pediatric population's needs.” The non-recognition or lack of recognition of the specialty of pediatric nursing in Spain may not always result from the information you have provided here. Do you have a scientific basis for this conclusion? Which levels of health were negatively affected?
We have reworded this section to avoid causal assertions not supported by direct data. Instead, we discuss the potential implications of poor integration of pediatric specialists, supported by scientific literature (e.g., Baird et al., 2016; Thibodeau et al., 2022; Morales Gil, 2023). We acknowledge the need for further research on measurable health outcomes related to specialization.
In general, I suggest that the discussion section be rewritten based on assumptions rather than definitive judgments, because if there is no scientific basis or health indicator, we cannot attribute the negative quality of health care only to the lack of recognition of pediatric nursing. It would also be nice to see what pediatric nursing will mean for the people of Spain in the future.
We fully agree. The discussion now avoids definitive statements, and future projections have been added.
Page 7- 4.5. Strengths of the Study
“The study meets the criteria of the SANRA Checklist (for narrative reviews), with a total item score of 12 [21].” How did you come up with the 12 points mentioned in this sentence? How many independent reviewers gave this score? I know that SANRA is used by editors and reviewers to set a standard, so it would not be appropriate to use it to describe the strengths of the work.
We have removed the sentence referencing the SANRA score. Instead, we describe the strength of using a mixed-methods and multilevel documentary review, including diverse data sources (academic, normative, institutional) and the structured thematic synthesis process.
All comments from Reviewer have been carefully addressed. We believe the revised manuscript is now clearer, more accessible to international readers, and better supported by evidence. Thank you very much.
Reviewer 2 Report
Comments and Suggestions for Authors
Thank you for the opportunity to review this manuscript on the evolution of pediatric nursing specialty in Spain. The manuscript adds important information because it identifies both inequities and inequalities as well as deficiencies in pediatric nursing care and training.
The manuscript is well-written and with interesting facts. However, some additional information would be helpful, especially for readers outside Spain/pediatrics.
- INTRODUCTION: Clarity could be shed on the topic (pediatric nursing) and on the Spanish context. For example, is the training integrated in the university system in Spain? Since the numbers per se do not illustrate the overall perspective, it would be interesting to know how many nurses there are in total in Spain, and how many pediatric nurses would be needed? Please consider adding some more information about pediatric nurses (I can see that this is later on evolved in Results), what they do, and why is this profession so important? How does all this position internationally?
- METHODS: A well-structured methods section.
- About 2.2 Documentary Search Strategy – please add the type of literature that was searched in 1. Academic Databases (all kinds of literature or just research articles?). How many articles were included (I did not find this information in Results).
- Did you use any statistical analysis software when calculating quantitative data?
- The ethics statement seems reasonable as there were no sensitive data or research persons involved.
- RESULTS: A well-structured results section with tables. However, I wonder about 3.1. The Importance of Pediatric Nursing Specialists, as it is not included in the current study aims. Maybe it could be moved to Introduction? Furthermore, the quotation ("The specialist nurse in pediatric nursing is a...") is quite lengthy, have you considered paraphrasing?
- DISCUSSION:
- You might want to start your discussion by briefly summing up the aims and the main results.
- Was there a particular reason for using subheadings? Please consider changing WHO's Perspective on Pediatric Nursing into something that reflects your findings, i.g. the state of pediatric nursing in Spain, as you want to discuss your findings in the light of WHO's Perspective on Pediatric Nursing rather than the other way round.
- Did you find any other possible explanations to the regional differences, e.g. economical, location of nursing schools, distribution of rural areas etc?
Author Response
Thank you for the opportunity to review this manuscript on the evolution of pediatric nursing specialty in Spain. The manuscript adds important information because it identifies both inequities and inequalities as well as deficiencies in pediatric nursing care and training.The manuscript is well-written and with interesting facts. However, some additional information would be helpful, especially for readers outside Spain/pediatrics.
We sincerely thank the reviewer for their insightful and encouraging comments. Your suggestions have contributed significantly to improving the clarity, structure, and international readability of the manuscript. Below we provide a point-by-point response to your recommendations, explaining how they have been addressed in the revised version.
- INTRODUCTION:
- Clarity could be shed on the topic (pediatric nursing) and on the Spanish context. For example, is the training integrated in the university system in Spain?
-
We have clarified that the training for the pediatric nursing specialty in Spain is part of the postgraduate residency-based system known as EIR (Enfermero Interno Residente), rather than part of the university curriculum. This is now clearly stated in the revised Introduction (pages 2–3).
- Since the numbers per se do not illustrate the overall perspective, it would be interesting to know how many nurses there are in total in Spain, and how many pediatric nurses would be needed?
-
We have added official data from the Spanish Ministry of Health, stating that in 2024 there were 264,219 registered nurses in Spain. Pediatric nurse specialists represent less than 2.5% of the total workforce.
We also added commentary on the OECD average of nurse density (2023), highlighting the national shortfall and the potential implication for pediatric care.We also discuss the gap between pediatric population needs and the current supply of specialists in both the Results and Discussion sections.
- Please consider adding some more information about pediatric nurses (I can see that this is later on evolved in Results), what they do, and why is this profession so important?
-
We have expanded the Introduction to include a summary of the official competencies of pediatric nurse specialists, as defined in Order SAS/1730/2010. These include:
- Developmental assessment
- Family support and education
- Technical and pharmacological interventions in pediatrics
- Promotion of child health
We also highlight international evidence of the benefits of specialized pediatric care. - How does all this position internationally?
-
We included a brief comparison with international trends. For example, Spain has 6.3 nurses per 1,000 inhabitants, compared to the EU average of 8.5. We cite OECD data and refer to WHO recommendations on specialization for child healthcare. Please see Introduction and Discussion.
- METHODS:
- A well-structured methods section.
-
Thank you for this positive assessment.
About 2.2 Documentary Search Strategy – please add the type of literature that was searched in 1. Academic Databases (all kinds of literature or just research articles?). -
We clarified in section 2.2 that we included:
- Peer-reviewed research articles
- Narrative reviews
- Policy documents
- Professional opinion papers
We selected documents based on their relevance and contribution to the understanding of pediatric nursing in Spain. - How many articles were included (I did not find this information in Results).
-
We now specify in material and methods section that the documentary review was carried out between January and March 2025 through a three-stage search strategy, following the PRISMA 2020 guidelines. Both the PRISMA 2020 flow diagram and the PRISMA checklist are available in the Supplementary Material. In results section, we now inform that the literature review yielded 20 academic and professional documents relevant to the development of the pediatric nursing specialty in Spain [5-9,13,14,16-27,29].
Thank you for this observation.
- Did you use any statistical analysis software when calculating quantitative data?
-
We have added a sentence to clarify that no statistical software was used. Quantitative analysis was descriptive, and calculations (percentages, frequencies) were done manually using Microsoft Excel.
- The ethics statement seems reasonable as there were no sensitive data or research persons involved.
-
Thank you. We have maintained the ethics statement, confirming that the study used only publicly available documentary sources and did not involve human subjects.
- RESULTS:
- A well-structured results section with tables. However, I wonder about 3.1. The Importance of Pediatric Nursing Specialists, as it is not included in the current study aims. Maybe it could be moved to Introduction?
-
We fully agree. This content has been relocated and integrated into the Introduction, where we discuss the role and relevance of the specialty. The Results now begin with the description of legal access routes to the specialty.
- Furthermore, the quotation ("The specialist nurse in pediatric nursing is a...") is quite lengthy, have you considered paraphrasing?
-
Yes, we have paraphrased the original quote to summarize its core content while still conveying its meaning. The revised sentence now enhances readability without losing accuracy.
- DISCUSSION:
- You might want to start your discussion by briefly summing up the aims and the main results.
-
The discussion now opens with a brief summary of the aim and key findings of the study, facilitating a smooth transition into the interpretive analysis.
- Was there a particular reason for using subheadings? Please consider changing WHO's Perspective on Pediatric Nursing into something that reflects your findings, i.g. the state of pediatric nursing in Spain, as you want to discuss your findings in the light of WHO's Perspective on Pediatric Nursing rather than the other way round.
-
As suggested, we have removed all subheadings in the Discussion and rewritten the section in a more narrative and cohesive style. The content formerly under “WHO's Perspective” has been integrated into a broader synthesis that discusses our findings in the light of global and national frameworks.
- Did you find any other possible explanations to the regional differences, e.g. economical, location of nursing schools, distribution of rural areas etc?
-
Yes. In the revised Discussion, we include new content pointing to possible explanatory factors for regional disparities, such as political will, funding priorities, uneven distribution of training centers, and rurality. These are acknowledged as potential contributors, alongside regulatory and structural factors.
All comments from Reviewer have been carefully addressed. We believe the revised manuscript is now clearer, more accessible to international readers, and better supported by evidence.
Round 2
Reviewer 1 Report
Comments and Suggestions for Authors
Dear Author(s),
I confirm that the authors have made improvements to the manuscript in line with my suggestions. This manuscript can be accepted after revision in accordance with the journal's editorial guidelines.